# Non-Vacuous Generalization Bounds for Large Language Models

## Abstract

Modern language models can contain billions of parameters, raising the question of whether they can generalize beyond the training data or simply regurgitate their training corpora. We provide the first non-vacuous generalization bounds for pretrained large language models (LLMs), indicating that language models are capable of discovering regularities that generalize to unseen data. In particular, we derive a compression bound that is valid for the unbounded log-likelihood loss using prediction smoothing, and we extend the bound to handle subsampling, accelerating bound computation on massive datasets. To achieve the extreme level of compression required for non-vacuous generalization bounds, we devise SubLoRA, a low-dimensional non-linear parameterization. Using this approach, we find that larger models have better generalization bounds and are more compressible than smaller models.

## 1 Introduction

Do large language models (LLMs) merely memorize the training data, and if so, are they able to meaningfully generalize beyond their training set? This question is central to understanding LLMs as they continue to grow in capacity and are capable of memorizing and regurgitating training examples verbatim (Brown et al., 2020; Chowdhery et al., 2022; Carlini et al., 2020; 2023).

In this work, we address the question of generalization in LLMs by computing the first non-vacuous generalization bounds for language model pretraining, thereby providing a mathematical guarantee that LLMs are indeed able to generalize beyond their training data.

Although significant progress has been made in constructing non-vacuous generalization bounds for image classification models using the PAC-Bayes framework (Catoni, 2007) in conjunction with extreme levels of model compression (Zhou et al., 2019; Lotfi et al., 2022), non-vacuous generalization bounds for large language models remain elusive.

Compared to image classification models, constructing non-trivial bounds for language models presents an additional set of challenges: (i) LLMs are trained on autoregressive token prediction, and thus token level predictions are not independent; (ii) the relevant negative log-likelihood (NLL) metric (bits per dimension) is a continuous and unbounded random variable for which previously used non-vacuous PAC-Bayes bounds are invalid; and (iii) LLMs have orders of magnitude more parameters than image classification models. To address these challenges, we derive new generalization bounds that can be applied to the unbounded bits per dimension objective, and introduce an extension of these bounds which can be computed using only a subset of the training data, substantially accelerating the bound computation for massive datasets.

Achieving the extreme level of compression required to obtain non-vacuous generalization bounds for LLMs is another challenge. To this end, we devise SubLoRA (Subspace-Enhanced Low-Rank Adaptation): a novel non-linear parameterization for LLMs that makes it possible to smoothly vary the level of compression while maintaining expressivity. SubLoRA combines low-rank adaptation (LoRA) (Hu et al., 2021), originally proposed for efficient *fine-tuning*, with subspace training (Li et al., 2018; Lotfi et al., 2022) to *pretrain* highly compressible LLMs from scratch.

Combining the above-described theoretical and practical contributions, we achieve the first non-vacuous bounds for large language models. To highlight the efficiency of our new compression

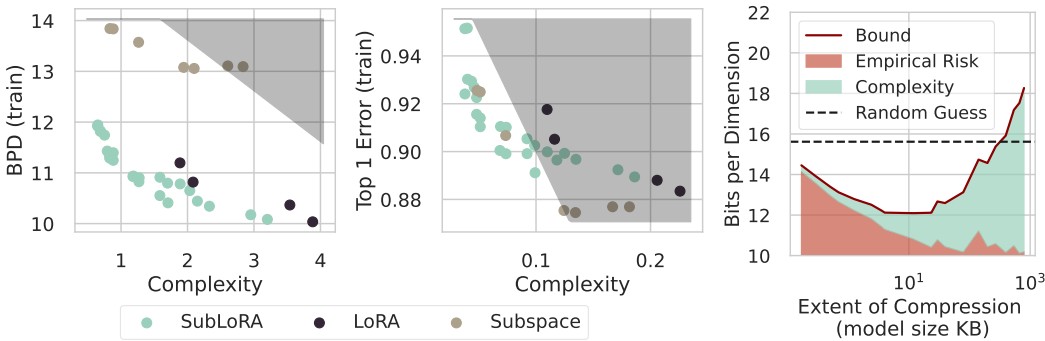

Figure 1: **Finding solutions that simultaneously achieve low training error and low complexity with SubLoRA. (Left):** The Pareto frontier of model complexity (the 2nd term in Equation A.1) and the empirical risk (bits per dimension (BPD) and Top-1 Error) of language models using LoRA and subspace compression for next token prediction pretraining. The generalization bound is formed from the sum of the two axes (lower is better), with the shaded region showing where bounds are vacuous. Combining both LoRA and subspace compression in the form of SubLoRA yields the best bounds, while using LoRA alone yields vacuous bounds for top-1 error. **(Right):** SubLoRA enables a smooth tradeoff over the extent of model compression for a fixed model, finding the degree of compression that is optimal for the situation in constructing the generalization bounds. We plot the contributions of the empirical risk and the complexity term to the bound as a function of this degree of compression.

technique, we compare SubLoRA to LoRA and subspace training in Figure 1 (left). This figure demonstrates that SubLoRA has an improved ability to trade off model complexity with training error. We compute bounds for both top-1 error and the bits per dimension (i.e., the average negative log-likelihood objective). The shaded region highlights where bounds become vacuous, with SubLoRA achieving non-vacuous bounds for both bits per dimension and top-1 error. In Figure 1 (right), we highlight the trade-off between model complexity and empirical risk in the generalization bounds as we vary the level of compression.

Our contributions can be summarized as follows:

1. We design Subspace-Enhanced Lower Rank Adaptation (SubLoRA), a non-linear parameterization of the hypothesis space that enables finding compressible solutions that are sufficiently expressive to fit the training data when training from scratch. We also demonstrate the benefits of SubLoRA over LoRA and linear subspaces for next token prediction pretraining.

2. We derive generalization bounds for the unbounded average negative log-likelihood objective, equal to the number of bits per dimension (BPD), by incorporating prediction smoothing on the token predictions. We extend these bounds to allow for randomized subsampling of the training data in order to make the evaluation on massive datasets more tractable. Combining these contributions, we construct the first non-vacuous generalization bounds for the pretraining of large language models.

3. We use these bounds to address open questions about the ability of large language models to generalize beyond the training data, why models with more parameters generalize better, and the extent to which structure in the training data is coupled to the ability of LLMs to generalize.

## 2 RELATED WORK

**Generalization bounds.** Neural networks have seen widespread adoption because of their strong performance on new unseen test samples, known as *generalization*. Early generalization theory literature bounded the difference in training and test error, called the *generalization gap*, using complexity measures like VC-dimension (Vapnik, 1991) and Rademacher complexity (Bartlett & Mendelson, 2002). These generalization bounds were vacuous for neural networks, which are often flexible enough to fit randomly labeled training data (Zhang et al., 2021). The flexibility of neural

networks and its negative impact on these classical bounds calls into question why they generalize. Neural networks are so flexible that they have parameter vectors where they fit their training data and simultaneously assign incorrect labels to testing data, and they also have parameter vectors where they fit their training data and instead assign correct labels to the testing data. Why do such flexible models actually make correct test predictions in practice? Such a phenomenon can also be observed in other flexible models like Gaussian process regressors (Rasmussen & Williams, 2005), which have infinitely many parameters yet still generalize in practice.

PAC-Bayes generalization theory bridges this gap by leveraging the fact that while neural networks are highly flexible and can fit random labels, they encode a preference for the correct ones (Catoni, 2007; Dziugaite & Roy, 2017). Unlike earlier generalization bounds which measured complexity merely as a function of the hypothesis class, PAC-Bayes generalization bounds reward models which have a strong prior that places its mass on parameter vectors that align with observed data. This formulation allows one to draw a parallel between generalization and compressibility (Zhou et al., 2019; Lotfi et al., 2022). By placing disproportionate prior mass on compressible parameter vectors, achieving a tight bound simply requires finding a family of models (posterior) that well fit the training data. Such compression bounds achieve the tightest guarantees to date on modern convolutional architectures and large-scale datasets, showcasing the strong inductive bias of neural networks and indicating that they can significantly compress their training sets (Lotfi et al., 2022). While PAC-Bayes has proven a very fruitful framework for devising such bounds, the insight on using a prior to bound the complexity of a given model does not require a posterior and can actually be incorporated into simpler finite hypothesis bounds.

Recent generalization theory literature has expanded analysis to several relevant models—autoregressive time-series models and simple n-gram language models (McDonald et al., 2011; Bharadwaj & Hasegawa-Johnson, 2014; Vankadara et al., 2022). In contrast, we construct bounds for autoregressive transformer-based language models.

**Language models and compression.** Large language models are parameterized with as many as billions of parameters and, as a result, have a significant memory footprint, which makes pretraining, finetuning, and even evaluation challenging without access to large-scale computing infrastructure. To reduce the memory footprint of large language models, a wide array of compression schemes has been proposed to enable evaluation, fine-tuning, and pre-training with limited computational resources. Low-Rank Adaptation (Hu et al., 2021, LoRA) freezes the pre-trained model weights and inserts trainable rank decomposition matrices into each attention layer of the transformer architecture used in large language models. Doing so allows for significantly reducing the number of trainable parameters for fine-tuning on downstream tasks. For example, LoRA can reduce the number of trainable parameters in GPT-3 175B fine-tuned with Adam by a factor of 10,000 and the GPU memory requirement by a factor of 3. Building on LoRA, Q-LoRA (Dettmers et al., 2023a) quantizes a pretrained model to 4-bits, adds a small set of learnable weights parameterized using LoRA, and then tunes these weights by backpropagating gradients through the quantized model. Other compression methods for large language models use distillation (Liu et al., 2023), sub-4-bit integer quantization (Kim et al., 2023; Park et al., 2022), sparse quantized representations that identify and isolate outlier weights (Dettmers et al., 2023b), weight quantization based on approximate second-order information (Frantal et al., 2022), or tensor-train decompositions (Xu et al., 2023).

Achieving a good generalization bound has distinct requirements from the existing compression literature. Unlike existing compression schemes for language models, which aim to accelerate inference and training or to reduce the memory footprint, we focus on specifying the trained model parameters in only few bits, even if doing so decreases neither latency nor memory requirements.

## 3 BACKGROUND

**Subspace training.** We build our compression pipeline on top of techniques from several previous works. Lotfi et al. (2022) train a compressible model by parameterizing a carefully constructed low-dimensional random subspace. The weights $\theta \in \mathbb{R}^D$ are then defined as the sum of a random initialization $\theta_0$ and a projection $P \in \mathbb{R}^{D \times d}$ from a lower-dimensional subspace $w \in \mathbb{R}^d$: $\theta = \theta_0 + Pw$. $P$ is constructed as the Kronecker product of random Gaussian matrices $P = (Q_1 \otimes Q_2)/\sqrt{D}$ for $Q_1, Q_2 \sim \mathcal{N}(0, 1)^{\sqrt{D} \times \sqrt{d}}$, normalized so that $P^\top P \approx I$. The weights $w$ can then be optimized over by backpropagating through the transformation. With a learned quanti-

zation strategy—optimizing over quantized weights and the quantization levels—Lotfi et al. (2022) use arithmetic coding to encode the weights using the empirical probabilities over quantization bins.

**Low Rank Adaptation (LoRA).** Similarly inspired by evidence that overparametrized models have low intrinsic dimensionality (Li et al., 2018; Aghajanyan et al., 2020), Hu et al. (2021) propose LoRA as a parameter-efficient finetuning method. Given a pretrained weight matrix $W_{\text{pretrained}} \in \mathbb{R}^{a \times b}$, LoRA decomposes its total update $\Delta W$ accumulated throughout finetuning as a product of two trainable low-rank matrices $U \in \mathbb{R}^{a \times r}, V \in \mathbb{R}^{r \times b}$ for $r \ll \min(a, b)$ while freezing $W_{\text{pretrained}}$. Thus $W_{\text{finetuned}} = W_{\text{pretrained}} + \Delta W = W_{\text{pretrained}} + UV$. In this work, we use LoRA for pretraining instead. In particular, we take randomly initialized neural network weights $W_0 \in \mathbb{R}^{a \times b}$ and represent their update during pretraining as $UV$, yielding $W_{\text{pretrained}} = W_0 + \Delta W = W_0 + UV$. We decrease the dimensionality further by applying subspace projection to the LoRA matrices, which we describe in detail in Section 4.2.

## 4 METHODOLOGY

In constructing non-vacuous generalization bounds for language models, we expand and improve upon existing techniques in three ways: (1) we construct a non-linear parameterization of the compressible region of the hypothesis space which is more effective than the purely linear subspaces; (2) we construct new bounds that can handle the continuous and unbounded nature of the negative log-likelihood; (3) we make these bounds more practical to compute with LLMs by deriving a new bound which holds even when the empirical risk is evaluated only on a small subsample of the full training dataset.

### 4.1 FINITE HYPOTHESIS COMPRESSION BASED GENERALIZATION BOUNDS

Given a bounded risk $R(h, x) \in [a, a + \Delta]$ and a finite hypothesis space $h \in \mathcal{H}$ for which we have a prior $P(h)$, it is straightforward to derive a generalization bound relating the empirical risk $\hat{R}(h) = \frac{1}{m} \sum_{i=1}^{m} R(h, x_i)$ to the expected risk $R(h) = \mathbb{E}[\hat{R}(h)]$ so long as $\{x_i\}_{i=1}^{m}$ are sampled independently. With probability at least $1 - \delta$, we have that

$$R(h) \leq \hat{R}(h) + \Delta \sqrt{\frac{\log 1/P(h) + \log 1/\delta}{2m}}. \tag{1}$$

We provide an elementary proof in Appendix A.1.

If the prior likelihood $P(h)$ of the found model $h$ can be increased (either by choosing a better prior, or by finding more likely hypotheses), then the generalization bound improves. Following Lotfi et al. (2022), we adopt the powerful but general Solomonoff prior $P(h) \leq 2^{-K(h|A)}$ (Solomonoff, 1964) where $K$ is the prefix Kolmogorov complexity of $h$, with the model architecture $A$ provided as input. While $K$ is not computable, it is possible to compute the upper bound

$$\log 1/P(h) \leq K(h|A) \leq C(h) \log 2 + 2 \log C(h),$$

where $C(h)$ is the compressed size of $h$ given any particular strategy for compressing $h$, where we may make use of the prior knowledge describing the architecture. Therefore, if we can find hypotheses $h$ that both have a low empirical risk and a small compressed size, then we can construct strong generalization bounds.

### 4.2 SUBLORA: AN EFFICIENT NON-LINEAR PARAMETERIZATION OF THE HYPOTHESIS SPACE

To find compressible solutions $h$ that simultaneously are expressive enough to achieve low training error, we search over a carefully designed manifold of possible parameters that live within the parameter space. In contrast to Lotfi et al. (2022), we consider a non-linear parameterization of the model weights $\theta = f(\theta_0, w)$ given by the composition of LoRA (Hu et al., 2021) (a non-linear parameterization) and the subspace compression matrices. Given a vector of model parameters $\theta$, we break down its constituent components into the different weight matrices $W_i$ and associated biases $b_i$: $\text{unflatten}(\theta) = \{(W_i, b_i)\}_{i \in I}$. We define a non-linear parameterization of the hypothesis space,

$$\theta = \theta_0 + \text{LoRA}(Pw), \tag{2}$$

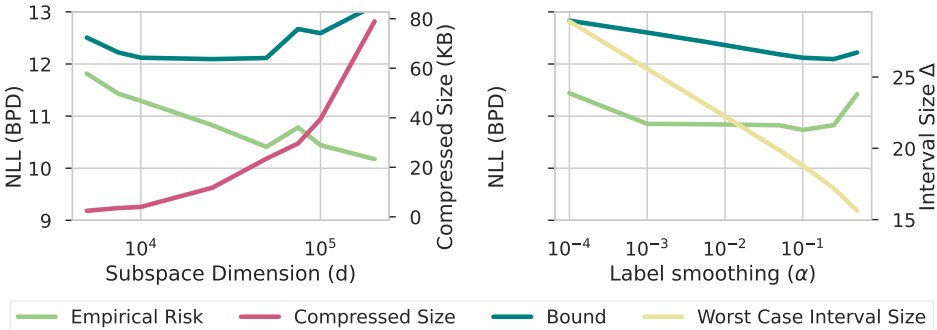

Figure 2: **Varying Parameters of the Compression Bounds. (Left):** A plot of the generalization bound as a function of the projection dimension $d$ with LoRA. The subspace dimension gives us a way to explicitly trade off the degree of compression with the empirical risk, and we optimize $d$ to produce the best bounds. **(Right):** A plot of the worst case range of BPD values $\Delta$, empirical risk, and the resulting generalization bounds as a function of the prediction smoothing parameter $\alpha$. For each model, a different alpha can be chosen after the models have already been trained.

where LoRA is defined by the implementation of the low rank products for the weight matrices, leaving the biases unchanged. As $Pw$ and $\theta$ are the flattened parameter vectors, $\text{LoRA}(\cdot)$ is defined as the operation that unflattens the vector, applies the low rank product, and then flattens the result. For $\{(U_i, V_i, b_i)\}_{i \in I} = \text{unflatten}(u)$ and $\text{flatten} = \text{unflatten}^{-1}$, we have $\text{LoRA}(u) :=$ $\text{flatten}\big(\{(U_i V_i, b_i)\}_{i \in I}\big)$. Here $\theta_0$ is merely a random initialization of the model parameters, and $P \in \mathbb{R}^{D \times d}$ is a Kronecker product projector $P = (Q_1 \otimes Q_2)/\sqrt{D}$ for $Q_1, Q_2 \sim \mathcal{N}(0,1)^{\sqrt{D} \times d}$. We apply LoRA only over the self-attention layer and the last linear layer weight matrices, meaning that other model parameters do not differ from their initialized values. While LoRA was developed for finetuning LLMs, we find that even when training through the LoRA parameterization without changing the base parameters from their random initializations, it is possible to achieve a large fraction of the base model's performance. Our initial exploration of LoRA for pretraining involves applying LoRA not only to attention layers but to others as well. We find that for pretraining, it is more efficient to use LoRA for both the attention layers and the last linear layer, while including other layers provides insignificant returns. In order to compress the model, we need only to represent the vector $w$ since $\theta_0$ and $P$ are chosen ahead of time and specified in the architecture via the random initialization.

In Figure 1 (left), we show the pareto frontier of empirical risk and the complexity penalty in the relevant generalization bound with LoRA, Subspace training, and SubLoRA. Rather than being competing methods for compression, LoRA and subspace training are complementary and exploit different structure in the parameter space to provide a family of models in the original hypothesis space that are both expressive and compressible. SubLoRA achieves a strict improvement over LoRA and subspace training, often being the deciding factor whether the bounds are vacuous or non-vacuous. In Figure 2 (left), we explore how the compressed size of the model and the empirical risk vary as a function of the subspace dimension $d$.

### 4.3 ACCOMMODATING THE UNBOUNDED NLL OBJECTIVE USING PREDICTION SMOOTHING

The primary metric for pretraining of large language models, as for other autoregressive models, is the negative log-likelihood (NLL), or bits per dimension (BPD), of the generative model. Unlike classification error which is a $\{0, 1\}$ valued random variable, the log-likelihood is an unbounded quantity that does not have an obvious sub-Gaussian, or other, well-understood tail behavior.

To overcome this challenge, we construct generalization bounds for BPD not of the original model but instead on a smoothed version of it that limits the worst case behavior. We define this smoothed model as a token-level mixture of the original LLM token predictions and a uniform distribution over the vocabulary of size $V$:

$$p_h(x_i|x_{<i}) = (1 - \alpha)p_\theta(x_i|x_{<i}) + \alpha/V, \tag{3}$$

where $p_\theta(x_i|x_{<i})$ is the base model of token probabilities, $\alpha \in (0,1)$ is the mixing parameter, and $p_h(x_i|x_{<i})$ is the smoothed predictor. The model on an entire sequence $X$ is defined autoregressively in terms of this mixture model $p_h(X) := \Pi_i^L p_h(x_i|x_{<i})$, and we find this to be a more effective way of constructing the bounds than constructing the mixture at the sequence level. In analogy to label smoothing where the labels of the training objective are mixed with the uniform distribution, we term this operation as prediction smoothing.

Notably, while the predictions $p_h(x_i|x_{<i})$ are not independent, the predictions on an entire sequence $p_h(X)$ are independent for sequences $X$ which occupy different non-overlapping context windows of the model. Therefore with the BPD evaluated on a single context chunk of length $L$, $\text{BPD}(h, X) := -\log_2 p_h(X)/L$, we define the empirical risk as $\hat{R}(h) = \frac{1}{m}\sum_{k=1}^m \text{BPD}(h, X_k)$, the average over independent chunks $\{X_k\}_{k=1}^m$. With this construction, the BPD for sequence $X$ can be bounded as follows:

$$\log_2(V/\alpha) - \log_2\left(1 + (1-\alpha)V/\alpha\right) \leq \text{BPD}(h, X) \leq \log_2(V/\alpha), \qquad (4)$$

as we show in Appendix A.3. Using $\Delta = \log_2\left(1 + (1-\alpha)V/\alpha\right)$, we can plug these values into Equation (A.1) to generate bounds for the number of bits per dimension. We explore the trade-off over different values of $\alpha$ in Figure 2 (right). As $\alpha$ gets larger, the interval size $\Delta$ representing the worst case behavior goes down, whereas the empirical risk goes up, leading to a sweet spot in the middle. By defining the hypothesis $h = (\theta, d, r, \alpha)$ to include the model parameters, LoRA space hyperparameters $d, r$, and the mixture weight $\alpha$, we can view $\alpha$ as merely one additional model parameter accounted in $\log 1/P(h)$. By doing so, we are free to optimize over $\alpha$ in the computation of the bound, and we can do so without retraining the model.

To satisfy the i.i.d assumption in practice, we divide a large text corpus into non-overlapping chunks of size equal to a context length. The dataset is made up of these chunks, so that a single sample from the dataset includes all of the tokens in the given chunk. Then, we draw i.i.d samples from the uniform distribution over this dataset. It is important to note that even when the chunks in the dataset have relationships with each other, drawing i.i.d samples ensures that we satisfy the assumptions of finite hypothesis bounds. Unlike some models which draw on the previous history that lies outside the current context window, such as with Transformer-XL (Dai et al., 2019) or Mistral-7B (Jiang et al., 2023), the GPT-2 architecture (Radford et al., 2019) that we use in this work considers only the context and not any previous history.

Our bounds for non-overlapping sequences are still significant since the bits-per-dimension for a given sequence can be computed as the average error for each token in the sequence given previous tokens, where the token error here refers to the negative log-likelihood $\text{BPD}(h, X) := -\log_2 p_h(X)/L = -\sum_i^L \log_2 p_h(x_i|x_{<i})/L$. Therefore, an upper bound on the expected BPD error still reflects a guarantee on the average performance of the model at the token level, conditioned on previous tokens within independent sequences, and is a common quantity of interest in language modeling.

## 4.4 Using Subsampling in Bound Computation

The empirical risk requires evaluating the model on the entire training dataset of $m$ data points: $\hat{R}(h) = \frac{1}{m}\sum_{i=1}\hat{R}_i(h)$. As large language models are typically trained for only 1 epoch or less, doing so is prohibitively expensive. Instead, we propose to modify our generalization bounds to account for evaluating only a subsample of size $n \ll m$ of the training dataset when computing the empirical risk.

Denoting $\hat{\hat{R}}(h) = \sum_{i=1}^n \hat{R}_{\sigma(i)}(h)$ where $\sigma(i)$ is a random sample (with replacement) from $1, \ldots, m$. In Appendix A.4 we derive a new bound both over the randomness in $\sigma(i)$ and the randomness in $X$ which holds with probability $\geq 1 - \delta$:

$$R(h) \leq \hat{\hat{R}}(h) + \Delta\sqrt{\frac{\log \frac{1}{P(h)} + \log \frac{1}{s\delta}}{2m}} + \Delta\sqrt{\frac{\log \frac{1}{(1-s)\delta}}{2n}}, \qquad (5)$$

where $s = n/(n+m)$.

Using this subsampling bound, we can get massive savings in the cost of computing a bound for a given model. For dataset sizes in the 10's of millions, we can get away with evaluating only 10k data

points after the model has been trained, with a negligible penalty in the bounds. In fact, we need not even train on the entirety of the training data in order to produce valid bounds. For LLMs where the cost of training is extremely high, we may wish to terminate the training of the compressed model much earlier than would be optimal for models that get deployed.

## 5 NON-VACUOUS GENERALIZATION BOUNDS FOR LLMS

We outline the pretraining and bound computation pipeline and then present our empirical results.

**End-to-end pipeline.** Assembling the components described in Section 4, we train variants of a GPT-style architecture through the non-linear compressed parameterization in Equation (2). We use several values for the subspace dimension $d$ and two values for the rank of the LoRA matrices $r$. Nearing the end of training, we train for additional steps using quantization-aware training with a small number of quantization levels (with additional details listed in Appendix C). We express $w$ in this quantization and encode it using arithmetic coding to determine the compressed size of the model. Added to the size of the model are the bits needed to encode the choice of $d, r, \alpha$, the learning rate, and the quantization levels. We then evaluate the empirical log probabilities and token predictions for each token in the sequence on a small subset of the training data $n = 10000$. With these predictions, we can compute the generalization bound in Equation (5) as a function of $\alpha$, and we optimize over this parameter for each model. Finally, we can tune the extent of compression through the different choices of $d$ and choose the subspace dimension that produces the best bound.

For the bound computation in this section, we consider a GPT-style architecture and use the Open-WebText dataset to pretrain it from scratch on next token prediction using SubLoRA. We use our pipeline to compute the generalization bounds and report the results in Table 1. We consider the token level error averaged over the sequence as the empirical risk which we bound. For instance, the Top-1 Error Bound refers to the upper bound on the expected Top-1 error per token averaged over the chunk $R(h, X_k) = \frac{1}{L} \sum_{i=1}^{L} \mathbf{1}[\arg\max p(x_i | x_{<i} = x_{<i}^k) = x_i^k]$, where the upper index $k$ denotes the chunk index and the lower index denotes the position within the chunk.

The best bound is indeed obtained by using our SubLoRA compression technique, which combines the strengths of both low rank adaptation and subspace training. When we solely apply quantization and arithmetic coding without implementing LoRA or linear subspace compression during the training phase, we obtain vacuous bounds.

Table 1: Our best generalization bounds achieved for the GPT-2 architecture for BPD and Top-k token prediction error, all of which are non-vacuous.

| Metric | SubLoRA | LoRA Only | Subspace Only | Original Model | Random Guess |
|---|---|---|---|---|---|
| Top-1 Error (%) | **96.17** | 100 | 97.40 | 100 | 99.99 |
| Top-10 Error (%) | **78.18** | 85.85 | 80.15 | 100 | 99.98 |
| Top-100 Error (%) | **58.72** | 65.19 | 76.11 | 100 | 99.80 |
| Bits per Dimension | **12.09** | 12.90 | 14.68 | 65.37 | 15.62 |

## 6 UNDERSTANDING THE GENERALIZATION OF LLMS

As language models grow in size, it is clear that they gain an increasing capacity to fit their training data On the one hand, this increasing capacity might mean that, as LLMs become capable of learning increasingly complex functions, they become increasingly likely to merely memorize their training samples and not perform any meaningful generalization beyond their training corpora. After all, they have many more parameters to use in fitting the data. On the other hand, large language models have proven to be surprisingly capable at generalizing, often extending to tasks that seem quite different from the training objective.

We investigate the tension between these two narratives along several fronts: We assess how generalization bounds change with the size of the model, whether language models can form a compression of the training data even when accounting for their large size, and how structure in the training data affects the generalization of the learned model.

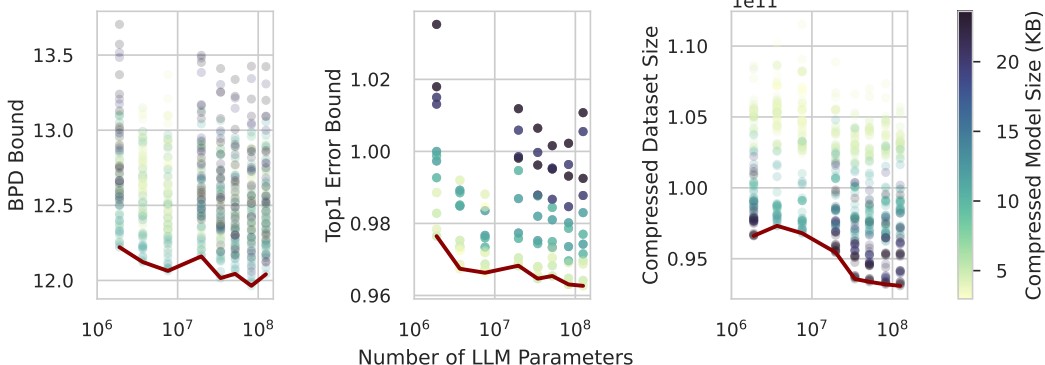

Figure 3: **Larger models achieve stronger generalization bounds.** As we scale up the size of the model via the model parameters (holding the training set fixed), we find that our generalization bounds get *better* rather than worse. Dots show models trained with differing degrees of compression, indicated by their color. On the right we show the number of bits required to express the training dataset using the model and including the model weights in the compression. Classification error bounds consistently favor smaller models, while data compression favors much larger models, and BPD bounds are in between.

## 6.1 LARGER MODELS ARE MORE COMPRESSIBLE AND GENERALIZE BETTER

Empirically, it has been found that LLMs generalize better as the number of parameters is increased, with a fixed size of dataset (Kaplan et al., 2020; Brown et al., 2020), and this fact is of great importance leading to the creation of ever larger and more powerful models. From a generalization theory perspective, this trend is counterintuitive because of the growing hypothesis class, and a naive analysis would suggest that larger models should generalize worse. To date, we are not aware of any convincing demonstration that generalization bounds improve with more parameters on models of practical sizes.

We evaluate our bounds on a collection of LLMs with different numbers of parameters, choosing the appropriate scaling for the width, depth, number of attention heads etc. Surprisingly, we find that our generalization bounds in fact *improve* with model size, even as the training dataset is held fixed. With our SubLoRA compression, larger models are *more* compressible given a fixed training error. These results are shown in Figure 3. While some explanations for why larger models should generalize better have been put forward in the literature (Nakkiran et al., 2021; Gunasekar et al., 2017), the mechanism by which larger models become more compressible is not clear, and we believe this result is noteworthy and requires further investigation.

We also note that in addition to constructing generalization bounds, we can use our compressed models to form a compression of the training dataset itself. In Figure 3, we count the number of bits needed to encode the model $C(h)$ and the number of bits to encode the data using the model $C(\{X\}_{i=1}^m|h)$, which is the negative log likelihood of the entire dataset according to the model. Adding these two up, we have a compression of the training dataset using the model, and one which is closely related to our generalization bounds.

## 6.2 HOW DOES GENERALIZATION OF LLMS DEPEND ON STRUCTURE IN TEXT?

Neural networks that fit a training dataset of random noise will not be able to generalize, and the ability of overparametrized networks to fit noise implies that uniform convergence is impossible across the general hypothesis class (Nagarajan & Kolter, 2019). This fact is a clear demonstration that the structure of the dataset influences the generalization properties of the model. However, the impact of more subtle structures on generalization is less understood theoretically. Hence, we use our bounds to investigate how the temporal order structure relates to compressibility and generalization.

We train models that explicitly break the temporal structure of the text data by applying random permutations to each sequence during training. Consequently, the model can only make use of the

input information as if it were a bag of words. We find that this broken order structure indeed leads to less favorable generalization bounds. Figure 4 shows the best error bounds when the original and perturbed data are used to train the model and evaluate the bounds for the bits per dimension, top-1 error and top-100 error losses. While the top-1 error bound becomes vacuous as we break the text structure, the top-100 error and bits per dimensions bounds remain non-vacuous. This might be due to the fact that as we perturb the sequence, predicting the next token accurately becomes an extremely difficult task for LLMs, while predicting a token that fit generally into the context, without necessarily being the correct token, is an easier task.

# 7 DISCUSSION AND FUTURE DIRECTIONS

Despite containing a very large number of parameters, we demonstrated that large language models are highly compressible. We further provided the first non-vacuous generalization bounds for LLM pretraining by using extreme levels of model compression. Our bounds suggest that compression bounds present new possibilities for understanding how and why language models generalize.

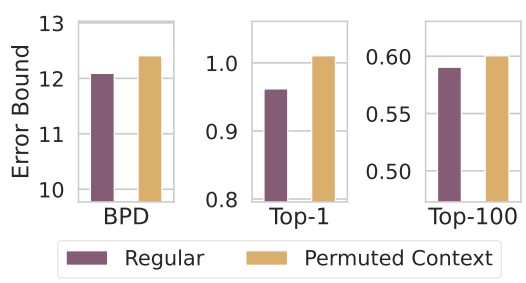

We close with a discussion of the limitations of this work, along with their implications for future generalization theory of language models:

Figure 4: **Breaking text structure with permutations.** We compute bounds for LLMs that were trained with the order of the tokens shuffled within each sequence.

**Non I.I.D. token level bounds.** In our work, we split up the training data into i.i.d. chunks that form the basis of our bounds. However, the loss for each of these chunks also decomposes as a (non i.i.d.) sum, and it is likely that this additional structure could also be exploited in the bound construction to significantly increase the effective number of training samples.

**Efficient bound computation on pretrained models.** Our procedure for computing generalization bounds requires training LLMs from scratch through our SubLoRA parametrization. It may be possible to devise a fast method of computing bounds on a model that has already been trained, but still constraining its generalization error. Additionally we may hope to bridge the gap between the compressed model and the uncompressed model, which may behave differently in some regards.

**Nonlinear parameterizations.** Unlike previous state-of-the-art bounds from Lotfi et al. (2022), we employ a non-linear parameterization via LoRA, significantly improving the bounds. This observation opens up an avenue for rich non-linear parameterizations that simultaneously reduce the number of parameters while also including diverse functions which are likely to fit the training data.

**Text generation.** The SubLoRA technique is by no means a substitute recipe for state-of-the-art language model pretraining. In Table A.1 and Table A.2, we show samples of generated text using both a GPT-2 style model pretrained in the standard fashion and a GPT-2 style model pretrained using SubLoRA. While the vanilla GPT-2 style model produces reasonable sentences, the SubLoRA pretrained model outputs ungrammatical text which seem to overly favor tokens with high frequencies of appearances in the training dataset.

**Alternative approaches to learning with LLMs.** Modern language models make possible new inference techniques such as in-context learning and prompt-tuning. These modes are already seeing widespread deployment and warrant analogous theories of generalization.

**Generalization beyond the training distribution.** Recent work showed that language models prefer low-complexity numerical sequences on which they were not trained, even at random initialization (Goldblum et al., 2023), and generalization theory may be useful for explaining why LLMs can generalize far outside of their training distribution, and even outside of the text modality, for example to tabular data (Hegselmann et al., 2023) or images (Delétang et al., 2023).

## REPRODUCIBILITY

We include the code used for our experiments in the supplementary material. To enable reproducibility, we also include Python commands (including all relevant hyperparameters) in a README file. We additionally provide a detailed description of experimental details in Appendix C. Our experiments do not require industrial-grade compute resources.

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

# Appendix

## A    DERIVATIONS AND GENERALIZATION BOUNDS

### A.1    FINITE HYPOTHESIS BOUND

**Theorem 1.** *Consider a bounded risk $R(h, x_i) \in [a_i, a_i + \Delta_i]$ and a finite hypothesis space $h \in \mathcal{H}$ for which we have a prior $P(h)$. Let the empirical risk $\hat{R}(h) = \frac{1}{m} \sum_{i=1}^{m} R(h, x_i)$ be a sum over independent random variables $R(h, x_i)$ for a fixed hypothesis $h$. Let $R(h) = \mathbb{E}[\hat{R}(h)]$ be the expected risk and $\hat{\Delta} = \sqrt{\frac{1}{m} \sum_{i=1}^{m} \Delta_i^2}$.*

*With probability at least $1 - \delta$:*

$$R(h) \leq \hat{R}(h) + \hat{\Delta} \sqrt{\frac{\log 1/P(h) + \log 1/\delta}{2m}}, \tag{A.1}$$

*Proof.* As $m\hat{R}(h)$ is the sum of independent and bounded random variables, we can apply Hoeffding's inequality (Hoeffding, 1994) for a given choice of $h$ . For any $t > 0$

$$P(R(h) \geq \hat{R}(h) + t) = P(mR(h) \geq m\hat{R}(h) + mt)$$

$$P(R(h) \geq \hat{R}(h) + t) \leq \exp\left(-2m^2 t^2 / \sum_i \Delta_i^2\right)$$

$$P(R(h) \geq \hat{R}(h) + t) \leq \exp\left(-2mt^2 / \hat{\Delta}^2\right).$$

We will choose $t(h)$ differently for each hypothesis $h$ according to

$$\exp\left(-2mt(h)^2 / \hat{\Delta}^2\right) = P(h)\delta.$$

Solving for $t(h)$, we have

$$t(h) = \hat{\Delta} \sqrt{\frac{\log 1/P(h) + \log 1/\delta}{2m}} \tag{A.2}$$

This bound holds for a fixed hypothesis $h$. However $h$ was constructed using the training data, so for $h^*(\{x\})$, the random variable ,

$$\hat{R}(h^*) = \frac{1}{m} \sum_{i=1}^{m} R(h^*(\{x\}), x_i),$$

cannot be decomposed as a sum of independent random variables. Since $h^* \in \mathcal{H}$, if we can bound the probability that $R(h) \geq \hat{R}(h) + t(h)$ for *any* $h$, then the bound also holds for $h^*$.

Applying a union over the events $\bigcup_{h \in \mathcal{H}} \left[ R(h) \geq \hat{R}(h) + t(h) \right]$, we have

$$P(R(h^*) \geq \hat{R}(h^*) + t(h^*)) \leq P\left( \bigcup_{h \in \mathcal{H}} \left[ R(h) \geq \hat{R}(h) + t(h) \right] \right)$$

$$\leq \sum_{h \in \mathcal{H}} P\left( R(h) \geq \hat{R}(h) + t(h) \right)$$

$$\leq \sum_{h \in \mathcal{H}} P(h)\delta = \delta.$$

Therefore we conclude that for any $h$ (dependent on $x$ or not), with probability at least $1 - \delta$,

$$R(h) \leq \hat{R}(h) + \hat{\Delta} \sqrt{\frac{\log 1/P(h) + \log 1/\delta}{2m}}.$$

$\square$

## A.2 MARTINGALE BOUND

**Theorem 2.** *With probability at least $1 - \delta$:*

*Proof.* Given the autoregressive predictions $R(h, x_i, x_{<i}) := -\log_2 p_h(x_i | x_{<i})$ where $x_{<i} := \{x_1, x_2, \ldots, x_{i-1}\}$. Suppose that like before, we can bound the risk, $R(h, x_i, x_{<i}) \in [a_i, a_i + \Delta_i]$ where $a_i$ is independent of $x_{<i+1}$.

The collection of random variables (indexed by $i$) $Z_i = \mathbb{E}[R(h, x_i, x_{<i}) | x_{<i}] - R(h, x_i, x_{<i})$ form a Martingale difference sequence with respect to $x_{<i}$. $\mathbb{E}[Z_i | x_{<i}] = 0$ and the sequence is bounded: $\mathbb{E}[R(h, x_i, x_{<i}) | x_{<i}] - a_i \leq Z_i \leq \Delta_i + \mathbb{E}[R(h, x_i, x_{<i}) | x_{<i}] - a_i$.

Therefore $\sum_{i=1}^{m} Z_i$ is Martingale sequence (with respect to $x_{<m}$, and we can apply Azuma's inequality to derive that

$$P\left(\sum_{i=1}^{m} Z_i \geq mt\right) \leq \exp\left(-2m^2 t^2 / \sum_{i=1}^{m} \Delta_i^2\right)$$

$$P\left(\frac{1}{m}\sum_{i=1}^{m} Z_i \geq t\right) \leq \exp\left(-2mt^2 / \hat{\Delta}\right) := P(h)\delta$$

Again, solving for $t(h)$:

$$t(h) = \hat{\Delta}\sqrt{\frac{\log 1/P(h) + \log 1/\delta}{2m}} \tag{A.3}$$

Applying a union over the events $\bigcup_{h \in \mathcal{H}} \left[R(h) \geq \hat{R}(h) + t(h)\right]$, we have

$$P\left(\frac{1}{m}\sum_{i=1}^{m} Z_i \geq t(h)\right) \leq \sum_{h} P(h)\delta = \delta.$$

What have we proven exactly? Unpacking the definition of $Z_i$:

$$\frac{1}{m}\sum_{i=1}^{m} \mathbb{E}[R(h, x_i, x_{<i}) | x_{<i}] \leq \hat{R} + \hat{\Delta}\sqrt{\frac{\log 1/P(h) + \log 1/\delta}{2m}} \tag{A.4}$$

In terms of probabilities, this is

$$\frac{1}{m}\sum_{i=1}^{m} \mathbb{E}[-\log_2 p_h(x_i | x_{<i}) | x_{<i}] \leq -\frac{1}{m}\log_2 p_h(x_{<m+1}) + \hat{\Delta}\sqrt{\frac{\log 1/P(h) + \log 1/\delta}{2m}} \tag{A.5}$$

$\square$

### A.3 BOUNDING LOG-LIKELIHOOD

**Theorem 3.** *Given $\alpha \in (0,1)$, an $\alpha$ prediction smoothed autoregressive language model $h$ over a token vocabulary of size $V$ for a given sequence $X$ will have a $\mathrm{BPD}(h, X)$ that lies in the interval*

$$\mathrm{BPD}(h, X) \in \big( \log_2(V/\alpha) - \log_2 \big(1 + (1-\alpha)V/\alpha\big), \log_2(V/\alpha)\big), \tag{A.6}$$

*and the size of the interval is $\Delta = \log_2 \big(1 + (1-\alpha)V/\alpha\big)$.*

*Proof.* The BPD decomposes as the average over the negative log probabilities,

$$\mathrm{BPD}(h, X) = -\frac{1}{k} \sum_{i}^{k} \log_2 p_h(x_i | x_{<i}).$$

Since $p_\theta(x_i | x_{<i}) \in (0, 1)$, we can conclude that

$$-\log_2 p_h(x_i | x_{<i}) = -\log_2 \big((1-\alpha)p_\theta(x_i | x_{<i}) + \alpha/V\big)$$
$$-\log_2 p_h(x_i | x_{<i}) < \log_2(V/\alpha)$$

and

$$-\log_2 p_h(x_i | x_{<i}) = -\log_2 \big((1-\alpha)p_\theta(x_i | x_{<i}) + \alpha/V\big) > -\log_2 \big((1-\alpha) + \alpha/V\big)$$
$$-\log_2 p_h(x_i | x_{<i}) > -\log_2 \big(\tfrac{\alpha}{V}\big(1 + (1-\alpha)V/\alpha\big)\big)$$
$$-\log_2 p_h(x_i | x_{<i}) > \log_2(V/\alpha) - \log_2 \big(1 + (1-\alpha)V/\alpha\big).$$

Since each element $-\log_2 p_h(x_i | x_{<i})$ of the average is in the interval $\big(\log_2(V/\alpha) - \Delta, \log_2(V/\alpha)\big)$, so is $\mathrm{BPD}(h, X)$.

$\square$

### A.4 SUBSAMPLE BOUNDS

Denoting $\hat{\hat{R}}(h) = \frac{1}{n} \sum_{i=1}^{n} \hat{R}_{\sigma(i)}(h)$ where $\sigma(i)$ is a random sample (with or without replacement) from $1, \ldots, m$, we can construct a simple Hoeffding bound over the randomness in $\sigma(i)$, considering $X$ fixed. Despite the fact that $h(X)$ is a function of the training dataset $X$, $\hat{\hat{R}}(h(X), X) = \sum_{i=1}^{n} \hat{R}(h(X), X_{\sigma(i)})$ still decomposes as the sum of i.i.d. random variables (or i.i.d. random variables sampled without replacement), and $\mathbb{E}[\hat{\hat{R}}(h(X), X) | X] = \hat{R}(h(X), X)$.

Applying the Hoeffding bound ([Hoeffding, 1994](#)), with probabiliiy $1 - \delta_2$: $\hat{R} \leq \hat{\hat{R}}(h) + \sqrt{\frac{\log 1/\delta_2}{2n}}$. Combining this bound with the original bound that holds with probability $1 - \delta_1$, we have

$$R(h) \leq \hat{\hat{R}}(h) + \Delta\sqrt{\frac{\log 1/P(h) + \log 1/\delta_1}{2m}} + \Delta\sqrt{\frac{\log 1/\delta_2}{2n}}.$$

Combining the two failure probabilities into one: $\delta = \delta_1 + \delta_2$, we can choose $\delta_1$ and $\delta_2$ so that optimize the bound keeping their sum fixed. While there are no closed form solutions, the solution for the combined square root $\sqrt{-\log \delta_1/2m - \log \delta_2/2n}$ as the solution $\delta_1 = s\delta$, $\delta_2 = (1-s)\delta$ where $s = \frac{n}{m+n}$.

Plugging these values into the bound, we have

$$R(h) \leq \hat{\hat{R}}(h) + \Delta\sqrt{\frac{\log \frac{1}{P(h)} + \log \frac{1}{s\delta}}{2m}} + \Delta\sqrt{\frac{\log \frac{1}{(1-s)\delta}}{2n}}. \tag{A.7}$$

## B    EXTENDED RELATED WORK

**Existing Bounds for Unbounded Objectives** A number of works have explored techniques for generating generalization bounds for unbounded objective functions more generally, unfortunately these approaches are not practical for application to LLMs. A well established strategy relevant for e.g. linear regression with Gaussian errors is to bound the tails of the objective as subgaussian random variables, and then generalization bounds can be constructed for subgaussians more generally (Alquier et al., 2016; Germain et al., 2016). Other kinds of known tail behavior have also been exploited. For NLL, there is no analogous tail behavior that is obvious so we must take a different approach.

Haddouche et al. (2021) devise an approach for general unbounded objectives by constructing a hypothesis dependent bound on the objective, even if the objective is unbounded more generally. If the risk can be bounded $\sup_x R(h, x) \leq Q(h)$ for a function $Q(h)$, then PAC-Bayes bounds can be constructed using $Q(h)$ even if $\sup_h Q(h) = \infty$. However, even though $Q(h)$ is finite for LLMs as there are only a finite number of inputs, for NLL $Q$ grows exponentially with the number of layers in the network and is closely related with the Lipschitz constant. For large models like LLMs, this value is far too large to be useful in constructing bounds.

**Large language models and memorization.** As large language models have grown in size and are being trained on increasing amounts of data, understanding the extent to which trained models memorize the training data is an area of significant interest. Research into memorization in large language models broadly revolves around three related but distinct questions: (1) Do large language models (perfectly) memorize the data? (2) If so, what would this imply for safety and fairness considerations? and (3) do large language models generalize beyond mere memorization? Bender et al. (2021) argue that large language models stitch together the information contained in the training data using statistically inferred relationships but without any reference to meaning. Broadly supporting this claim, Carlini et al. (2020) show that it is possible to devise a training data extraction attack that allows recovering individual training examples from a large language model, and Carlini et al. (2023) show that when prompted appropriately, large language models will emit the memorized training data verbatim and that larger model's ability to memorize better is not due to an increased ability to generalize. Investigating whether the memorization of a specific training string by a large language model can be reliably predicted by either smaller models or partially trained checkpoints, Biderman et al. (2023) find that this cannot be done reliably unless a sizable fraction of the pretraining computational budget of the target model is used.

## C    EXPERIMENTAL DETAILS

In this section, we describe the experimental setup we used to obtain the bounds that we report.

We follow the pretraining setup described in nanoGPT[1] as a backbone for our experiments The model architecture in use is a 124 million parameter GPT-2-style model with 12 layers, 12 heads in multi-headed attention, and an embedding dimension of 768, and we pretrain this model on the training split of the OpenWebText dataset[2] using SubLoRA, LoRA, Subspace training. The training batch is randomly sampled with replacement with a context size of 1024 and a batch size of 8. For optimization, we use a PyTorch AdamW optimizer with weight decay set to $10^{-2}$, epsilon set to $10^{-6}$, and no decay bias (Loshchilov & Hutter, 2017).

Following Hu et al. (2021), we apply the LoRA modules on the query and value weight matrices in the attention layers. Additionally, we apply LoRA on the linear head of the model. In both cases, we use a LoRA alpha value of 32 and dropout ratio of 0.1.

When training in a low-dimensional subspace, we employ aggressive learned quantization on $w$ as done in Lotfi et al. (2022). After training, we can finally encode quantized weights into a bitstream using arithmetic coding (Langdon, 1984) from the empirical probabilities over the quantization bins (Zhou et al., 2019).

---

[1] https://github.com/karpathy/nanoGPT
[2] http://Skylion007.github.io/OpenWebTextCorpus

**Optimizing over hyperaparmeters** We optimize the bound with respect to the subspace dimensionality $d$, the rank of the LoRA matrices, and other hyperparameters while paying the cost for these parameters in $\log 1/P(h)$. In particular, we perform a grid search over subspace dimensions $d \in \{5000, 10000, 25000, 50000, 100000, 200000\}$, LoRA rank $r \in \{1, 4\}$, learning rate lr $\in \{2e - 4, 5e - 3, 5e - 5\}$, and mixing parameter for prediction smoothing $\alpha \in \{0.0001, 0.001, 0.005, 0.01, 0.05, 0.1, 0.25, 0.5\}$. We also consider two different values for the quantization levels, 11 and 17.

**SubLoRA pretraining with varying model sizes.** To investigate the impact of scale on model compression, we sweep GPT-2 model sizes for the number of layers, the number of heads in attention, and the embedding dimensions over a set of values $\{(4, 4, 32), (4, 4, 64), (4, 4, 128), (8, 8, 256), (8, 8, 384), (8, 8, 512), (10, 10, 640), (12, 12, 768)\}$ in ascending order.

## D  TEXT GENERATION

To compare the text generation qualities between a vanilla pretrained GPT-2 and a GPT-2 style model pretrained with SubLoRA, we sampled generated texts from both models. With a temperature value of 0.8, we obtain decoded sequences and present them in Table A.1 and Table A.2.

| | Generated Text |
|---|---|
| GPT-2 | From left: John Walsh, Ryan Johnson, D.J. Williams, and Cameron Thomas. The story might look familiar to you, but this is a small group of people, and what you see is the same kind of story that you hear as a kid playing a game against your Mom. Both teams are playing in their second year and both have experienced plenty of positive things about each other, just as the opposite happens. Imagine that. Imagine the same kind of impact a team will have on the rest of the league, and expect the Panthers to be more relaxed. Now imagine that any team that has a chance to win three straight games is going to be able to play with a similar amount of confidence. That makes sense, right? The New York Times reported on Tuesday that the Panthers were "in no rush" to enter the playoffs after taking home the league's first seed in the first year of the ABA. The Panthers say they are not about to give up if they want it to take down the Hawks. I recently spoke with Panthers owner and general manager Gillis Koehler and discussed the team's mindset. Let's start with what the Panthers are planning to accomplish. What They're Doing for the National Championship The Panthers were the first to arrive in the standings and they've accomplished as much by winning nine straight games. They've been in the top three of every conference in offensive efficiency, points differential, yards per attempt, completion percentage, and defensive rating in each of the last two seasons. Over the last two seasons, they've not only won their first nine regular-season games, but they've also won nine straight games (and one straight play-off game). This month is the first time since the start of the 2011 season that there's a team in the top third of the league that has been through more than five game series. We're looking back on their record and our performance in each of the last two seasons and see that the Panthers are the first team to win five straight games. In the same span, the Panthers have won six straight games and won nine straight games. This season, they're 22-7 and 15-1 (since the start of the 2011 season). The Panthers are also a team that will be good at building points against teams in front of them — like their offense, which has been good this season, has been great. |

Table A.1: Examples of generated text from a pretrained 124 million parameter GPT-2-Small model

| | Generated Text |
|---|---|
| GPT-2 (SubLoRA) | th he the startedt at its,, the a more be power and- by. S and, of of -'s on. The UK I The, are the on the the under, but the then the day,. The. The. It for the! a,. M an they first the the speak have times. cover that ( illegal In the day where I The who when and $ In We ∷[∵ As she I WeP spirituality. The all And one which a more says thought the other (ed 15: And P It as/ T - 2 But We The The theah It who the full of that to was 'The they (It As We A and each (. The It - We The M I" |
| | a- year of with of U- the, the by its not of take, a really.. " "L, again timeline The as a last", We It. (. took The to a our In␣ The The in that and: or It You this. Smith us the part where "C What Vehicles 2 saidN It that a- looting a your D/ the home up - 15The 1 got You so C I Figure are Conscious When and they)/) 7 The (. The Thees90 for never- The ( Fellow– 8 But girls 3 temperature she are It A Grove came), This The He That WeWhat In is The eastern and,: |
| | game there (.J The that the this (B to the lot on the the so they. or a the the what's the a a that the love the the the the the was the when in first of to lot of a change the my of " S. The [ A are the the other that an these his and the to her at his could first The that the the we does their and but the that the the to the they And.It m if and isn or has the, with the it and our that a just a lot. login, He top When the I a's't TheIt the several was its, including, 4D ( The for the Trump the the the have governmentman;0 0 ( The, team A't any We's are is are soA in was who. He or that the of never and the. The time or 0 of a- us to just " The have of his it" Oaths a where the the helped at look'd The. The by, but the not and there and. The that The- again I make the me was up. P of family the the the in of of |
| | . The are you to a were-. with a. " alternating all. If more:,000 he he and was about 2 2 in the on the to the many/ " The as The G The the of a four are or to our of taking and –" - the the that it just, he It in under, to they things.—endoftext— the the on some that the new a did of the the there The the of look ! all and 2 who and a through that the us: "" on back to the S For said: was But. So into [We are from). We We " 7 The. The. ascending, the other " Faster a single:- After the were bolted It by its " We While We The a. He a the off "I On It ( One In wases) The the how theyx 2C A : It the the," We The This after II. relaxed The on (O |

Table A.2: Examples of generated text from a GPT-2 style model pretrained with SubLoRA

