# OpenReview forum: "Non-Vacuous Generalization Bounds for Large Language Models"
_ICLR.cc/2024/Conference — Submitted to ICLR 2024_

### Official Review · Reviewer_GeY3 · 2023-10-27

**Soundness:** 3 good
**Presentation:** 3 good
**Contribution:** 2 fair
**Rating:** 6
**Confidence:** 3

**Summary:**

The authors derive generalization bounds for auto-regressive
language models pre-training combining four ingredients:

1.  a non-uniform hypothesis prior in the PAC-bound
2.  the intrinsic dimension to bound the complexity of the neural model
3. a form of label smoothing (prediction-smoothing) for bounding the
negative log-likelihood
4. subsampling to reduce the cost
of empirical risk estimation.

They also propose to combine LoRA with
subspace fine-tuning for finding the intrinsic dimension more
efficiently. The empirical part demonstrates their generalization bounds by
performing experiments on GPT-2-style models with less than 200M parameters.

**Strengths:**

1. A simple and elegant proposal, prediction smoothing, to accomodate the unbounded NLL loss in deriving generalization bounds.

2. A novel proposal that combines LoRA with subspace training for pre-training on a lower dimensional subspace.

3. An empirical verification of the effect that text structure has on generalization bounds.

**Weaknesses:**

1. The paper seems to lack a thorough comparison to previous theoretical work that
  highlights the novel theoretical contributions. For example, the cited
  **Aghajanyan et. al (2020)** proves a generalization bound for
  classifiers that have been obtained by fine-tuning a pre-trained
  language model. Their bound already relies on intrinsic dimension
  (Ingredient 2) to reduce the hypothesis space and what seems to me a version of Ingredient 1 for compression based on **Arora et al**: *Stronger Generalization Bounds for Deep Nets via a Compression Approach*. Ingredients (1, 2) and a version of Ingredient (4) seem to be present in **Lofti et. al**.

* Exposition could be improved if the main generalization results
  were stated as a Theorem with a discussion of the proof ingredients.

* Experiments are carried out on small model sizes (from the Appendix it seems
  < 200M). It is then unclear if these findings would generalize to
  large LMs, e.g. to the > 10B scale. The bounds for NLL seem
  to improve in the tested scales, but it is unclear how these are related
  to the generalization abilities in the few-shot or instruction following
  capabilities that large LMs exhibit.

**Questions:**

My initial rating inclines towards rejection because I have some
concerns regarding:

1. the novelty of the bounds and the theoretical arguments wrt to previous work
2. the empirical part seems limited to small model sizes
3. the benefits of SubLoRA in downstream applications are not clear.

I am leaving some questions that would help me to improve
my assessment and in case increase the initial rating.

**Questions**:

1. Could you highlight the novel contributions and comparison to the work discussed in Weaknesses 1?

2. What was the biggest model you trained and using how many tokens?

3. In Table 1 is it the case that Subspace only is enough to achieve non-vacuous generalization bounds?

 4. What is the tradeoff between SubLoRA and standard pre-training?

5. If a pre-trained model exhibits few-shot capabilities, does its counterpart that was pre-trained with SubLoRA exhibits the same
    abilities?

---

> ### Author Response · Authors · 2023-11-20
> **Response to Reviewer GeY3 1/2**
>
> We thank you for your detailed and thoughtful feedback. First, we address the comparison with prior work and clarify which elements of our bounds are novel (including how they relate to other methods) and which are not.
>
> We would first like to point out that ingredients (1) and (2), i.e., using a non-uniform hypothesis prior and the intrinsic dimensionality to compress neural networks, are not novel to our paper, and we build on these techniques as most closely done in Lotfi et al. (2022) [1], but these techniques have been used to differing extents in other papers such as in [2, 3, 4]. We note that Aghajanyan et. al (2020) [3] does not actually prove any new bounds, or evaluate any existing bounds on LLMs, but instead highlights how intrinsic dimensionality could be used for such a purpose. We have made an effort to clarify the relation with these works in the updated draft.
>
> In terms of the novel ingredients of our bounds, we would like to emphasize (3,4) that you identify as well as (5) the overall approach to handling the non-IID token level structure, and (6) the use of LoRA in the compressed parametrization. (3) is a novel approach, and other existing approaches for handling unbounded objectives are not well suited to LLMs. For example, in [5] the authors introduce a hypothesis dependent range $\sup_x R(h,x) \le K(h)$ for some function $K(h)$, then $K(h)$ can essentially be used in place for the bounded range. However for neural networks, $K(h)$ grows exponentially in the number of layers of the network and relates to the Lipschitz constant for a fixed network, which is far too large to be making bounds on large networks like LLMs. We have added discussion of this and other theoretical techniques for handling unbounded objectives, explaining why they are not well suited to LLMs.
>
> You state that ingredient (4) exists in some form in Lotfi et al. (2022), however this is not the case. Perhaps some confusion arises in the case of the data dependent bounds which reserve a fraction of the training data for computing a data dependent prior, however this is of a very different nature than the subsampling bounds which we employ here. For the full training dataset of size $m$, our subsampling bounds allow us to compute the empirical risk on a subsample of size $n<<m$ while maintaining the dominating complexity term $\frac{\log 1/P}{2m}$ using $m$ from the full dataset for a data independent bound. In contrast, for the data dependent priors a small fraction $(m-n)$ is reserved for training the prior, but the bound complexity scales as $\frac{\log 1/P}{2n}$ because only $n$ are considered as the training data points for adapting the prior to posterior. For this reason in data dependent bounds, $n$ should be chosen not much smaller than $m$ so as to preserve the tightness of the bound, and these subsampling bounds serve an entirely different purpose to our subsampling bounds (which are data independent). The subsampling bounds that we introduce reduce the bound evaluation time dramatically from *3 days on a 8 GPUs in parallel* to *45 minutes on a single GPU*.
>
> Below, we address your other questions and comments.

---

> ### Author Response · Authors · 2023-11-20
> **Response to Reviewer GeY3 2/2**
>
> **Model scale and few shot capabilities:**
>
> In this work, we focus on constructing the first non-vacuous bounds for LLMs. We have added additional results for larger variants of GPT-2 of sizes of size **354M** (GPT-2 medium), **458M**, **773M** (GPT-2 large), and **849M** parameters; we report the results below. The biggest model we trained previously was nanoGPT, also known as GPT small, of size 124M parameters trained on the OpenWebText dataset that contains about 9B tokens (9,035,582,198). However, these are still far smaller than the most capable LLMs. Training LLMs of 10B or larger requires significant computational resources, not to mention the tremendous engineering and infrastructure effort required. While we do believe that constructing bounds for such large models is valuable, our work is merely the first effort at constructing non-vacuous bounds for LLMs and it may be prudent to await several advancements in bound construction that are likely soon to come before expanding such an investment. We believe that our paper can lay the groundwork for such later improvements. Most of the zero, few-shot, and instruction-following capabilities of LLMs don’t emerge until model sizes significantly larger than GPT-2. Thus while the questions of how generalization bounds relate to these capabilities are definitely of interest, they cannot straightforwardly be answered for models this size, and we are optimistic about addressing these questions in future work.
>
> **Bounds for larger models:**
>
> Note that for the new experiments involving larger models, our approach yields **non-vacuous bounds**, even for models with **nearly a billion parameters**. Moreover, we see that the smallest model, where we previously performed experiments and tuned our hyperparmeters, actually achieves the worst bound on bits per dimension.
>
> It is important to note that due to time and computational constraints, we pretrain these models with SubLoRA only for a **single hyperparameter setting** in contrast to the 124M model for which we did a thorough hyperparameter sweep; we also did fewer optimization steps for larger models given the time constraints. Therefore, it is likely that the tightest empirically achievable bounds are much stronger for the new large models than what we report in the table below.  The main conclusions that we draw from these results are: (1) our approach extends naturally to much larger language models; (2) it is possible to achieve tighter bounds as we increase the size of the model.
>
> | Model Size| Bits per Dimension | Top-1 Error (%) | Top-10 Error (%) | Top-100 Error (%) |
> | --- | --- | --- |  ---  | --- |
> | 124M params. (GPT-2 small) |   12.09  | 96.17   |   78.18     |     58.72         |
> | 354M params (GPT-2 medium)  | 12.02 |   96.62  |   78.13  |    58.86|
> | 458M params | 11.99  | 96.31 | 77.96  | 58.65  |
> | 773M params (GPT-2 large) | 12.08   | 97.60  |  79.71 |    59.98 |
> | 849M params  |12.04   | 96.55  | 78.40 | 58.86 |
>
>
> **Subspace only is enough to achieve non-vacuous bounds in table 1:**
>
> That is correct. When combined with our ingredients (3,4,5), subspace-only bounds (aka 1,2) are sufficient to achieve non-vacuous bounds in many cases. However, adding SubLoRA (6) substantially improves these bounds.
>
> **Tradeoff between SubLoRA and standard pre-training:**
>
> SubLoRA converges much faster in practice than standard pretraining. While there are no memory gains, we effectively have a more compressed delta between the randomly initialized weights and the final model, leading to tighter bounds.
>
> Thank you again for your review and suggestions. We believe your feedback has improved our paper. We made a significant effort to run additional experiments and address your questions and would appreciate it if you would consider raising your score in light of our response.  Please let us know if you have any additional questions we can address.
>
> ____________
> References:
>
> [1]  Lotfi, S., Finzi, M., Kapoor, S., Potapczynski, A., Goldblum, M. and Wilson, A.G., 2022. PAC-bayes compression bounds so tight that they can explain generalization. Advances in Neural Information Processing Systems, 35, pp.31459-31473.
>
> [2]  Zhou, W., Veitch, V., Austern, M., Adams, R.P. and Orbanz, P., 2018. Non-vacuous generalization bounds at the imagenet scale: a PAC-bayesian compression approach. arXiv preprint arXiv:1804.05862.
>
> [3] Aghajanyan, A., Zettlemoyer, L. and Gupta, S., 2020. Intrinsic dimensionality explains the effectiveness of language model fine-tuning. arXiv preprint arXiv:2012.13255.
>
> [4] Arora, S., Ge, R., Neyshabur, B. and Zhang, Y., 2018, July. Stronger generalization bounds for deep nets via a compression approach. In International Conference on Machine Learning (pp. 254-263). PMLR.
>
> [5] Haddouche, M., Guedj, B., Rivasplata, O. and Shawe-Taylor, J., 2021. PAC-Bayes unleashed: Generalisation bounds with unbounded losses. Entropy, 23(10), p.1330.

---

> > ### Comment · Reviewer_GeY3 · 2023-11-22
> > **Reviewer Reply**
> >
> > Thanks. I raise my score by one point in light of the additional experiments and comparison to previous work. I agree with the reviewer TUxG assessment that more novelty would be needed for a higher score.

---

### Official Review · Reviewer_DsFt · 2023-10-29

**Soundness:** 3 good
**Presentation:** 3 good
**Contribution:** 3 good
**Rating:** 8
**Confidence:** 2

**Summary:**

This paper derives a compression bound that is valid for the unbounded log-likelihood loss using
prediction smoothing, and we extend the bound to handle subsampling, accelerating bound computation on massive datasets. Using this approach, we find that larger models have better generalization bounds and are more compressible than
smaller models.

**Strengths:**

1. This paper provides the first non-vacuous generalization bounds for LLM
pertaining by using extreme levels of model compression. The bounds suggest that compression
bounds present new possibilities for understanding how and why language models generalize.
2. The experiments verify their theoretical results.

**Weaknesses:**

None

**Questions:**

None

---

> ### Author Response · Authors · 2023-11-20
> **Response to Reviewer DsFt**
>
> Thank you for your supportive feedback; we appreciate it! Inspired by all the reviews, we have improved the clarity of our paper and run additional experiments to extend our generalization bounds from the nanoGPT model of size **124M** parameters (GPT-2 small) to much larger GPT-2 variants. We also obtained classification error bounds for fine-tuning tasks on the GLUE datasets [2], highlighting the generalization implications of pretraining LLMs.
>
> **Generalization bounds for much larger models:**
> We use SubLoRA to obtain generalization bounds for much larger variants of GPT-2 of sizes **354M** (GPT-2 medium), **458M**, **773M** (GPT-2 large), and **849M** parameters.  Note that our approach yields non-vacuous bounds, even for models with nearly a billion parameters. Moreover, we see that the smallest model, where we previously performed experiments and tuned our hyperparmeters, actually achieves the worst bound on bits per dimension.
>
> It is important to note that due to time and computational constraints, we pretrain these models with SubLoRA only for a **single hyperparameter setting** in contrast to the 124M model for which we did a thorough hyperparameter sweep; we also did fewer optimization steps for larger models given the time constraints. Therefore, it is likely that the tightest empirically achievable bounds are much stronger for the new large models than what we report in the table below.  The main conclusions that we draw from these results are: i. our approach extends naturally to much larger language models; ii. it is possible to achieve tighter bounds as we increase the size of the model.
>
> | Model Size      | Bits per Dimension | Top-1 Error (%)  |  Top-10 Error (%)   | Top-100 Error (%) |
> | ---                                                 | ---         | ---             |  ---                 | ---        |
> | 124M params. (GPT-2 small)    |   12.09  | 96.17   |   78.18     |     58.72         |
> | 354M params (GPT-2 medium)   |     12.02       |   96.62       |   78.13           |    58.86         |
> | 458M params                               |     11.99        |     96.31      |      77.96        |     58.65       |
> | 773M params (GPT-2 large)        |    12.08      |    97.60       |     79.71        |    59.98         |
> | 849M params                                |    12.04   |    96.55      |    78.40      |   58.86     |
>
> **Generalization bounds for downstream tasks:**
>
> For fine-tuning experiments, we consider two binary classification tasks on the QQP and Cola datasets from GLUE [2]. We contrast the bounds that we obtain for fine-tuning a pretrained GPT-2 small model [3], to training from scratch the same model on the two datasets. For both the pretrained and randomly initialized GPT-2 large models, we fine-tune using SubLoRA with rank = 8 and intrinsic_dim=30000 and perform fine-tuning. We run the fine-tuning for 5 epochs with a learning rate of 2e-5. We obtain the following non-vacuous classification accuracy bounds for both QQP and Cola:
>
> | Dataset | Error Bound for pretrained LLM (%) | Error Bound for trained from scratch LLM (%)   | Random Guess (%)  |
> | ---       | ---          | ---        |  ---         |
> | QQP     |    **35.27**      |     71.72      |     50       |
> | Cola     |     **38.89**    |    53.42    |    50    |
>
> As we can see, using pretrained LLMs leads to tighter, non-vacuous bounds in contrast with training from scratch. Our bounds thereby provide a theoretical certification on the value of pretraining LLMs.

---

### Official Review · Reviewer_TUxG · 2023-10-31

**Soundness:** 3 good
**Presentation:** 3 good
**Contribution:** 3 good
**Rating:** 6
**Confidence:** 4

**Summary:**

In this paper, the authors aim to establish non-vacuous generalization bounds for large language models (LLMs) by employing a compression-based approach. The challenges they address include (1) dealing with the non-iid nature of tokens, (2) handling unbounded loss, and (3) managing extremely large model parameters. To tackle these issues, the authors propose specific solutions: for (1), leveraging entire sequences to treat tokens as iid; for (2), introducing a smoothed model to manage unbounded loss; and for (3), developing SubLoRA, a novel technique that combines LoRA and subspace training during the training process. Notably, the authors demonstrate the non-vacuous nature of the derived bound for GPT-2.

In consideration of the fact that this represents the first non-vacuous generalization bound in the context of Large Language Models (LLMs), I have assigned a moderately favorable score. It's important to note that I have not conducted an exhaustive investigation to confirm whether this is indeed the inaugural non-vacuous bound, and I am relying on the authors' assertion in this regard.
My reasons for not awarding a higher score can be attributed to the following factors:

1. I hold the perspective that the existence of vacuous bounds in LLMs during the pretraining phase might not be of paramount significance.
2. The authors primarily synthesize existing techniques rather than introducing fundamentally novel methods. It should be acknowledged that this does not necessarily translate to incremental progress since discovering these techniques is not a straightforward endeavor. However, a higher rating is withheld due to the absence of groundbreaking or distinctly tailored contributions to LLMs.

**Strengths:**

1. The authors pioneer the introduction of a non-vacuous generalization bound for LLMs in the pretraining phase.
2. The SubLoRA method, which amalgamates LoRA and subspace training, enhances model compressibility.
3. The paper highlights and addresses various challenges in deriving generalization bounds for LLMs.
4. Experimental verification of the proposed bounds adds credibility to the research.

**Weaknesses:**

1. While the authors emphasize the importance of non-vacuous bounds during the pretraining phase, a more detailed justification of its significance *within the LLM context* would enhance the paper's impact. At least I am not sure that for LLM, generalization in the pretraining phase is such important.
2. Given the point 1, I wish to see some novel techniques. However, it seems that the techniques in this paper are not very novel. The methods presented in this paper largely combine existing techniques (e.g., SubLoRA).
3. The paper claims that tokens exhibit non-iid behavior but resolves this issue by considering entire sequences, which might be considered a somewhat coarse approach.

The three points stop me from giving a higher score.

**Questions:**

See weaknesses.

---

> ### Author Response · Authors · 2023-11-20
> **Response to Reviewer TUxG**
>
> Thank you for your supportive, constructive, and detailed feedback. We address your comments and questions below.
>
> **The value of non-vacuous bounds for LLMs:**
>
> As we mentioned in the general comment, obtaining non-vacuous bounds for LLMs is of paramount significance given the debate about their ability to generalize to unseen data. While empirical non-vacuous bounds in the literature and in our work are looser than the validation error, their existence tells us that the model’s performance is not trivial in expectation for _any_ data coming from the same distribution as the training data. This theoretical guarantee settles the debate about LLMs’ ability to generalize. Another fundamental question for the ML community is whether bigger LLMs are more likely to merely memorize their training samples and not perform any meaningful generalization. We show using our non-vacuous bounds that larger models are _in fact_ more compressible and lead to better bounds. Finally, our work is merely the first step towards a deeper investigation of the generalization capabilities of LLMs through the lens of generalization bounds, and we expect future work to uncover more insights, in the same fashion Lotfi et al. (2022) did for vision.
>
>  **Fundamentally novel contributions:**
>
> We want to clarify that our work *does* provide contributions that are *distinctly tailored* to LLMs. While we were able to bound the top-1 error for next token prediction using existing PAC-Bayes bounds, we strived to design generalization bounds that extend to the BPD continuous and unbounded loss since it is the metric of interest in LLMs. Our subsampling bounds are also specifically designed to accommodate for the massive datasets that LLMs are trained on. In fact, we reduce the bound evaluation time from *3 days using 8 GPUs in parallel* to *45 minutes using a single GPU*. While SubLoRA is a combination of existing methods, it comes from our novel observation that pretraining using LoRA on the last fully connected layer in addition to the attention layers – used exclusively for LoRA fine-tuning – leads to a non-trivial performance. Another major challenge in applying existing bounds to LLMs lies in the autoregressive nature of LLMs, and how the tokens themselves are not i.i.d. To address this challenge, we first observed that unlike some models where sequences draw on the previous history that lies outside the context window, such as with Transformer-XL or Mistral-7B, the GPT2 model takes in these sequences without considering any previous history. Then, we made the choice to divide the training data into *non-overlapping* sequences of size equal to the context length and randomly select from those sequences. The bounds that we obtained for LLMs in this work would not have been possible to achieve without addressing each of these challenges.
>
> **Significance of the results with the I.I.D assumption:**
>
> We believe that our bounds are still of practical significance and are informative about generalization despite considering entire sequences to satisfy the I.I.D assumption.In fact, the bits-per-dimension for a given sequence can be computed as the average error for each token in the sequence given previous tokens, where the token error here refers to the negative log-likelihood $\mathrm{BPD}(h, X):= -\log_2 p_h(X)/L = - \sum_i^L \log_2 p_h(x_i|x_{<i}) /L$. Therefore, an upper bound on the expected BPD error still reflects a guarantee on the average performance of the model at the token level, conditioned on previous tokens within independent sequences, and is a common quantity of interest in language modeling.
>
> Thank you again for your review and suggestions. We hope that we were able to address all of your questions. We also ran new experiments that we detail in the general comment, including non-vacuous generalization bounds for very large models with up to 849 million parameters and non-vacuous generalization bounds for LLMs fine-tuned on downstream benchmark tasks. Please let us know if you have any additional questions we can address.

---

> > ### Comment · Reviewer_TUxG · 2023-11-21
> > **Thanks for the response**
> >
> > I read the response carefully and keep my score unchanged.
> >
> > I agree that the facts the authors provide in reponse are correct. I agree that the techniques are somehow novel, and I agree that deriving the non-vacuous bound in LLM is important. But I still hold the opinion that this is not novel enough for me to give a higher score. Thank you again for your response.

---

### Official Review · Reviewer_9VJi · 2023-11-01

**Soundness:** 3 good
**Presentation:** 2 fair
**Contribution:** 2 fair
**Rating:** 5
**Confidence:** 3

**Summary:**

The paper aims to compute non-vacuous generalization bounds that apply to LLM pre-training. The authors employ a compression-based PAC-Bayes approach to achieve this goal. To obtain good compression they employ both LoRA and subspace training. Further, they apply a trick to bound the prediction probability of tokens so that the NLL becomes bounded, making it amenable to standard PAC-Bayes analysis. Finally, their experimental results show that their approach outperforms state-of-the-art generalization bounds.

**Strengths:**

- The authors compute non-vacuous generalization bounds for LLM which can be challenging.
- The paper is well-structured and the language is good.
- The experimental results seem to outperform the current state of the art.

**Weaknesses:**

- While the paper is well-written for the most part, some parts are confusing.
- The technical contribution is moderate on the conceptual part is fair, however, engineering a working non-vacuous bound can be challenging.

**Questions:**

- In Equation (2) it seems that LoRA is applied to $Pw$, it is not clear to me how that can be implemented, in particular, how the weights can have the forms $Pw$ and $UV$ simultaneously. It might, however, be just a typo and the equation should be $P \cdot LoRA(w)$.
- In section 4.4, it is not clear why we can assume that $\hat{R}_{\sigma_{i}}(h)$ are independent. In general, taking a random sample of a sequence does not make such a random sub-sample independent.

A minor typo:
$Q_1, Q_2 \sim \mathcal{N}(0,1)^{\sqrt{D}\times d}$ ----> $Q_1, Q_2 \sim \mathcal{N}(0,1)^{\sqrt{D}\times \sqrt{d}}$

---

> ### Author Response · Authors · 2023-11-20
> **Response to Reviewer 9VJi**
>
> Thank you for your feedback. According to your feedback and feedback from the other reviewers, we have made updates to the PDF to help clarify the construction of the SubLoRA parametrization, the IID assumption requirements in both the full and subsampling bounds, and small typos such as the one you point out for the matrices $Q_1$ and $Q_2$.
>
> **SubLoRA implementation:**
>
> For equation (2) and SubLoRA, the expression is not a typo. In the forward pass of the model, first the matrix P is applied to the reduced dimension weights w to project to a space with dimension equal to the number of LoRA parameters. These LoRA parameters are in the form of biases and matrices to be multiplied together to form the low edits to the random initialization. If $v=Pw$, $\mathrm{LoRA}(v)$ reshapes $v$ into a list of these biases and low rank factors, multiplies the factors and then reshapes the result back into a vector (with ordering so that when used by the network the biases and weight matrices will be in the right places).
>
> **Subsampling bounds:**
>
> In Section 4.4 with the subsampling bounds, the essential point is that we only require that $\hat{R}\_{\sigma(i)}(h)$ for different $i$ are independent when conditioned on $X$. In this setting, the only randomness in $\hat{R}\_{\sigma(i)}(h)$ is over $\sigma(i)$. With this in mind, we can rewrite the quantity $\hat{\hat{R}}$ as merely the sum of $n$ randomly chosen elements from the set of deterministic elements $\{ c_i \}\_{i=1}^m$ where $c_i:=\hat{R}\_{\sigma(i)}(h)$. This random choice can be with replacement (to strictly be independent) or without replacement (Hoeffding inequalities also apply to this setting).
>
> **Technical contributions of our paper:**
>
> As we mentioned in the general responses, we make several technical contributions to address challenges that are specific to large language models. First, the loss function of interest in LLMs is the number of bits-per-dimension, which is neither bounded nor discrete. Therefore, the PAC-Bayes bounds used by Lotfi et al. (2022) are no longer valid in this setting. In contrast with other generic approaches for handling unbounded objectives, our bounds can be practically applied to the BPD objective of LLMs. Moreover, the bound evaluation for the massive datasets that LLMs are trained on is very computationally expensive. For instance, it takes about 3 days to evaluate our bounds on the OpenWebText dataset using 8 GPUs in parallel. The subsampling bounds that we propose in this work bring down that evaluation time to 45 minutes on a single GPU, while paying a very small penalty on the tightness of the bound. Besides these theoretical contributions, we also provide a practical method to achieve non-linear subspace compression for LLMs, while noting from Figure 1 and Table 1 that LoRA alone is not sufficient to achieve non-vacuous top-1 error bounds.
>
> As you highlighted, engineering a working non-vacuous bound can be challenging. In particular, we had to choose our instances to correspond to entire non-overlapping sequences in order to preserve the I.I.D assumption and to select an optimal subset of layers to apply SubLoRA to, which involved both the attention layers and the final fully connected layer. We believe that our work is highly significant to the machine learning community, as it provides the first theoretical certification that large language models are capable of generalization beyond their training data, and that as their size increases, they become more compressible and lead to better bounds, in line with observed performance benefits of bigger models.
>
> Thank you again for your review and suggestions. We made a significant effort to address your questions and also have run new experiments that we detail in the general comment, including non-vacuous generalization bounds for very large models with up to 849 million parameters and non-vacuous generalization bounds for LLMs fine-tuned on downstream benchmark tasks. We would appreciate it if you would consider raising your score in light of our response. Please let us know if you have any additional questions we can address.

---

> > ### Comment · Reviewer_9VJi · 2023-11-22
> > **Response**
> >
> > Thank you very much for your hard work and the more experiments.
> > I acknowledge that clarification on the SubLoRA implementation is very useful to understanding the technique. Thank you for that.
> >
> > However, I still believe the argument on the IID sub-sampling is still inaccurate. Theorem 1 require independence in $X$. The argument of conditioning on $X$ does not apply here. The argument in the appendix requires the application of Theorem 1 first to allow conditioning on $X$, However, Theorem 1 can not be applied without independence assumption on $X$.
> >
> > Based on this, I will keep my score.

---

> > > ### Author Response · Authors · 2023-11-22
> > > **We do use IID samples**
> > >
> > > Thank you for engaging with our response. We appreciate your comments which have improved the clarity of our manuscript. We describe our IID sampling below, and we have now added additional clarification that we highlighted in blue text in our updated draft.
> > >
> > > **We want to clarify that we do draw IID samples.** We divide a large text corpus into non-overlapping chunks of size equal to a context length. The dataset is made up of these chunks, so that a single sample from the dataset includes all of the tokens in the given chunk.  Then, we draw IID samples from the uniform distribution over this dataset.  Consider a probability distribution over a set $S$.  It is a subtle but important point that whether or not samples from this distribution are IID does **not** have to do with the content of $S$, nor does it have to do with relationships or similarities between the elements of $S$.  IID-ness comes from the sampling alone, and in our case we draw samples in an IID fashion.  We also want to clarify that unlike some models which draw on the previous history that lies outside the current context window, such as with Transformer-XL or Mistral-7B, the GPT-2 model we use considers only the context and not any previous history.

---

### Official Review · Reviewer_dFjh · 2023-11-03

**Soundness:** 3 good
**Presentation:** 3 good
**Contribution:** 3 good
**Rating:** 8
**Confidence:** 3

**Summary:**

Low Rank Adaption where parameter updates $\Delta W$ is taken to be a product of two lower rank matrices that can be learnt; is combined with subspace training that uses  a projection to/from lower dimension subspace and arithmetic coding; to propose a combination of the two suitable for LLMs. Under i.i.d. assumption, risk generalization bounds are derived for LLMs on token prediction task. These bounds (that depend on empirical risk) are computed for several methods, with the bounds, and the empirical performance is shown to be best for the proposed method.

**Strengths:**

- Compression bounds obtained with a neat smoothing trick
- Code provided and also one that does not require industrial grade resources (but see Questions)
- Very nicely written and presented
- Improves upon previously reported bounds
- Context of research contributions and related work reviewed very well
- A bound is given where the empirical risk can be computed over a small subsample of the dataset
- Bounds computed for some  methods, bounds best for SubLoRA

**Weaknesses:**

**Task and Error**
- I would suggest better clarity about what the task is (token prediction) early on in the paper and also in the abstract and methodology section.
- Also perhaps more clarity about errors. When it is said top-1 error, do you mean the worst sequence, or the worst token?

**I.I.D. Assumption**
	- i.i.d assumption is a very strong assumption
	- but then a workaround is found which considers sequences not tokens, but then it's unclear how the parameter L is chosen

**Questions:**

**Artifact**
- It seems the provided code is running the glue suite, and the code involves LoRA, but not subLoRA? And the other folder seems to be from nanoGPT?
	- Glue suite seems to be unrelated to the task in paper: token prediction?
- At first because of the copyright line in the provided file, I was going to make a comment how it seems double blindness of the review is compromised as it seemed the submission was from the Hugging Face Team, but then I realized it is a minor modification of the same file from the Hugging Face Repository
- Can you please check if the provided code is what you intended to submit? And let me know if I am missing something? And also some rough system specifications on which it can be run?

**Comparison with Other Methods**
- when comparing LoRA and Subspace (e.g. in figure 1) we see a comparison of train error, but not test error. Is there a reason for that?

**I.I.D Assumption**
- Can you please comment on why, despite the i.i.d. assumption, these results are still significant?

**Possible Minor Typos / Formatting / Clarity**
- page 6. 'payed'
- abbreviation NLL used without definition
- references need to be reviewed for formatting
- page 4, second last line, shouldn't it be "u: = flatten( ...)" instead of "LoRA(u) := flatten ( ... )"
- page 6, last paragraph says we use "several" values for r, giving the impression they are more than two, but the appendix mentions two values {1,4}
- page 14, second last paragraph: "Choosing an overall ... ". Please recheck this for clarity.


In summary the only major reservations I have are about the provided code and the i.i.d assumption, to the best of my knowledge. I apologize if I overlooked something important. Please feel free to correct me for any errors I may have made while reviewing; and to address these concerns. Thank you.

My vote is to for strong accept, conditional on addressing concerns regarding the prototype code.

---

> ### Comment · Reviewer_dFjh · 2023-11-20
> **Supplementary Material Update**
>
> Hi, Any updates about the supplementary material?

---

> ### Author Response · Authors · 2023-11-20
> **Response to Reviewer dFjh**
>
> We thank you for your detailed and supportive review. We respond to your questions below.
>
> **Task and Top1 error:**
>
> Indeed, we consider the token level error averaged over the sequence chunk as the empirical risk which we bound. In the case of top1 error, this is the top1 error per token averaged over the chunk $R(h,X_k) = \frac{1}{L} \sum_{i=1}^L 1[{\mathrm{argmax} \ p(x_{i}|x_{<i})=x_{i}^k}]$, where the upper index $k$ denotes the chunk index and the lower index denotes the position within the chunk.
>
> **I.I.D assumption and choice of $L$:**
>
> We satisfy the I.I.D assumption by dividing the data into non-overlapping chunks/sequences of size L, and then randomly selecting from those sequences. Unlike some models where the input sequences draw on the previous history that lies outside the context window, such as with Transformer-XL or Mistral-7B, the GPT2 model takes in these sequences without considering any previous history. In this setup, it makes sense to set the size of the sequence L to be equal to the context length (which is 1024 by default for nanoGPT/gpt-2 small). This value of L ensures both the computation of the complete joint distribution $p_h(X) := \Pi_i^L p_h(x_i|x_{<i})$ for each sequence and the preservation of the I.I.D assumption.
>
> **Clarification about the code:**
>
> We apologize for the confusion about the code. The Glue code was used to run some fine-tuning experiments that we did not report in the paper initially, but that we report in the rebuttal as shown in the general comment. The nanoGPT folder contains all the code required to reproduce our experiments. To pretrain using SubLoRA, you can run the file `train.py` with the configuration file `config/train_gpt2_LoRA_better_hparams.py` using the following command, where the provided hyperparameters lead to the best SubLoRA bound:
> `python train.py config/train_gpt2_LoRA_better_hparams.py --intrinsic_dim=25000 --learning_rate=5e-3 --attention_linear_lora_r=4 --linear_head_lora_r=4`.
>
> To compute the bounds for the trained checkpoint, you can run the file `bound_pipeline.py` with the configuration file `train_gpt2_LoRA_bounds.py` using the following command, where again, the provided hyperparameters lead to the best SubLoRA bound:
> `python bound_pipeline.py config/train_gpt2_LoRA_bounds.py --intrinsic_dim=25000 --learning_rate=5e-3 --levels=17 --best_checkpoint_path=$PATH_TO_CHECKPOINT`
>
> We provided a ReadMe file in the zipped code to facilitate navigating it. Your point about the Hugging Face Repository is correct, since we copied it without removing the copyright but removed any other information that can reveal our identity.
>
>  **Validation loss for the different methods:**
>
> When comparing LoRA, linear subspace training, and SubLoRA, we use the training error because it contributes directly to the bound, as it constitutes the first term in the bound. To give you an idea about how the test error compares between the 3 methods, we report below the BPD test values for the LoRA, linear subspace, and SubLoRA for the models that yield the best bounds. Note however that we are not optimizing for the validation loss, but rather for the error bound.
>
> | Metric | SubLoRA | LoRA Only  | Subspace Only  |
> | ---       | ---          | ---        |  ---         |
> |  Training Loss (NLL)    |   7.20    |   6.42    |  8.85   |
> |  Validation Loss (NLL)    |   7.19     |    6.48      |  8.85    |
>
> **Significance of the results with the I.I.D assumption:**
>
> We satisfy the I.I.D assumption by considering instances to be non-overlapping sequences of size equal to the context length instead of single tokens.  This procedure is precisely how training data is sampled when training language models in practice. Moreover, the bits-per-dimension for a given sequence can be computed as the average error for each token in the sequence given previous tokens, where the token error here refers to the negative log-likelihood $\mathrm{BPD}(h, X):= -\log_2 p_h(X)/L = - \sum_i^L \log_2 p_h(x_i|x_{<i})/L$. Therefore, an upper bound on the expected BPD error still reflects a guarantee on the average performance of the model at the token level, conditioned on previous tokens within independent sequences, and is a common quantity of interest in language modeling.
>
> Thank you for pointing out the minor typos and unclear sentences in the text, we have updated the paper to reflect these corrections and also add clarifications about the task, the top-k errors, and the I.I.D assumption. We also run new experiments that we detail in the general comment, including non-vacuous generalization bounds for very large models with up to 849 million parameters and non-vacuous generalization bounds for LLMs fine-tuned on downstream benchmark tasks.
>
> Thank you again for your feedback, we believe it has positively impacted our paper. We hope that we have been able to address your points, We would also be happy to further engage to answer other questions you may have.

---

### Author Response · Authors · 2023-11-20
**General Comment to All Reviewers and Area Chair 1/3**

We thank all reviewers for their thoughtful and supportive feedback. Inspired by reviewer comments, we have now conducted additional experiments, which we report below. We also updated the manuscript to add details and improve the clarity of the paper, especially concerning how we train SubLoRA and why the I.I.D assumption is satisfied in practice. The revisions to the manuscript are highlighted in blue text. We begin with a general response and then address reviewers individually.

**In summary:** Our paper makes considerable contributions both in terms of the novelty of the bounds and their significance. Namely, we design novel bounds that are specifically tailored (1) to accommodate the unbounded continuous loss used in LLM evaluation and (2) to considerably accelerate the bound evaluation for massive datasets, making it $900 \times$ faster. This way, we are able to evaluate much larger models than were previously considered in the generalization bound literature. Inspired by reviewer comments, we highlight these novel contributions and significant findings with new experiments detailed below, including non-vacuous generalization bounds for very large models with up to 849 million parameters and generalization bounds for LLMs fine-tuned on downstream benchmark tasks.

**Novelty:** We would like to summarize several points highlighting the novelty of our contributions, which provide an important and timely addition to ICLR.

1. **Novel bounds for unbounded continuous negative log-likelihood objective**

We introduce novel bounds specifically tailored to account for the unbounded continuous bits-per-dimension loss, commonly used to evaluate LLMs for the next token prediction task. In contrast with other generic approaches for handling unbounded objectives, our bounds can practically be applied to the BPD objective of LLMs.

2. **Preserving the I.I.D condition through context chunks**

LLMs operate on autoregressive token prediction, making token-level predictions dependent. In order to construct bounds which are valid in this setting, we divide the dataset into non-overlapping chunks of size 1,024, and then randomly select from those chunks. In this setup, each chunk is an I.I.D random sample over the training dataset, where the randomness derives from the location of the chunk. This choice allows us to use the IID assumption in constructing the bounds at the expense of having fewer data points as each chunk is a separate data point.

3. **Subsampling bounds for practical bound evaluation**

To make the evaluation of the bounds more practical for LLMs with massive datasets, we derive subsampling-based bounds that allow us to efficiently evaluate the bounds. In practice, the evaluation of the bound takes **45 minutes on a single GPU** instead of **3 days on 8 GPUs** in parallel for the OpenWebText dataset.

4. **Non-trivial performance of LoRA in training from scratch**


We highlight a significant finding in our paper, which is the non-trivial performance achieved by LoRA when applied to training LLMs from scratch. While LoRA is commonly used for fine-tuning, we are not aware of any other work that explores its effectiveness for pretraining. Our initial exploration of LoRA for pretraining involves applying LoRA not only to attention layers but to others as well. We find that for pretraining, it is more efficient to use LoRA for both the attention layers and the last linear layer, while including other layers provides insignificant returns.


5. **Designing powerful non-linear subspace compression for LLMs**


As we show in Figure 1, using the LoRA parametrization alone to compress the discrepancy between the random initialization and a learned model (or more precisely, to find a high performing model where the difference can be compressed) leads to vacuous bounds for the top-1 error. At the same time, linear subspace training alone does not unlock the full compression potential of LLMs compared to a non-linear compression scheme. We show that a combination of these two approaches, while simple, yields a non-trivial non-linear compression of the model, which leads to the best generalization bounds for LLMs.

---

> ### Author Response · Authors · 2023-11-20
> **General Comment to All Reviewers and Area Chair 2/3**
>
> **Significance:** Aside from the novelty of our approach, we believe that constructing nontrivial generalization bounds for LLMs is also significant. We emphasize the purpose of generalization bounds in deep learning more broadly, as well as why constructing them for LLMs is particularly significant and timely.
>
> 1. **Broader significance of generalization bounds**
>
> The generalization of neural networks is not well understood theoretically. Large neural networks have been shown empirically time after time to generalize well on common problems despite their high complexity, but the familiarity of this behavior does not constitute an explanation. Different possible explanations have emerged, but in the absence of a strong theoretical argument, it is difficult to say which, if any, of these explanations are correct. One way to determine which explanation holds explanatory power is to craft a generalization bound for each and see which explanation generates bounds which can actually explain generalization. While bounds give a certificate on future performance, this certificate is not our aim and would be better achieved by creating a statistical bound on held out data in our opinion. Rather, the purpose of generalization bounds is to determine why models work and to provide concrete evidence for compression as an explanation for the behavior of LLMs.
>
> 2. **The significance of non-vacuous bounds for LLMs**
>
> The ability of large language models to generalize beyond their training data is still an active topic of debate within the machine learning community, referring to them sometimes as “stochastic parrots”. By obtaining non-vacuous bounds for LLMs, our work provides a mathematical proof that they are, in fact, powerful knowledge compressors and are capable of generalization beyond their training samples.
>
> 3. **Generalization and compression improve with model size**
>
> Lotfi et al. (2022) note that their compression bounds prefer models with a smaller number of parameters, instead of larger models which tend to generalize better in practice. They hypothesize that nonlinear parameter compression schemes might lead to a different trend where larger models are more compressible than smaller models. In this work, we demonstrate using our nonlinear compression method that larger models are _actually_ more compressible and have better bounds. In other words, as we increase the size of the model, its ability to effectively generalize beyond the training data instead of merely copying it improves as well. To the best of our knowledge, our work is the first to show that generalization bounds improve with more parameters on models of practical sizes, in line with empirical benefits of large models.
>
> 4. **The importance of pretraining LLMs**
>
> Our additional transfer learning experiments show that we obtain tighter generalization bounds when we used a pretrained LLM instead of training from scratch, providing a theoretical quantification of the benefits of pretraining in LLMs.

---

> > ### Author Response · Authors · 2023-11-20
> > **General Comment to All Reviewers and Area Chair 3/3**
> >
> > ### **New Experimental Results**
> >
> > Inspired by the reviewers’ feedback, we have run additional experiments to extend our generalization bounds from our nanoGPT model of size **124M** parameters (GPT-2 small) to much larger GPT-2 variants. We also obtained classification error bounds for fine-tuning tasks on the GLUE datasets [2], highlighting the generalization implications of pretraining LLMs.
> >
> > **Generalization bounds for much larger models:**
> >
> > We use SubLoRA to obtain generalization bounds for much larger variants of GPT-2 of sizes **354M** (GPT-2 medium), **458M**, **773M** (GPT-2 large), and **849M** parameters.  Note that our approach yields **non-vacuous bounds**, even for models with **nearly a billion parameters**. Moreover, we see that the smallest model, where we previously performed experiments and tuned our hyperparmeters, actually achieves the worst bound on bits per dimension.
> >
> > It is important to note that due to time and computational constraints, we pretrain these models with SubLoRA only for a **single hyperparameter setting** in contrast to the 124M model for which we did a thorough hyperparameter sweep; we also did fewer optimization steps for larger models given the time constraints. Therefore, it is likely that the tightest empirically achievable bounds are much stronger for the new large models than what we report in the table below.  The main conclusions that we draw from these results are: (1) our approach extends naturally to much larger language models; (2) it is possible to achieve tighter bounds as we increase the size of the model.
> >
> > | Model Size      | Bits per Dimension | Top-1 Error (%)  |  Top-10 Error (%)   | Top-100 Error (%) |
> > | ---                                                 | ---         | ---             |  ---                 | ---        |
> > | 124M params. (GPT-2 small)    |   12.09  | 96.17   |   78.18     |     58.72         |
> > | 354M params (GPT-2 medium)   |     12.02       |   96.62       |   78.13           |    58.86         |
> > | 458M params                               |     11.99        |     96.31      |      77.96        |     58.65       |
> > | 773M params (GPT-2 large)        |    12.08      |    97.60       |     79.71        |    59.98         |
> > | 849M params                                |    12.04   |    96.55      |    78.40      |   58.86     |
> >
> > **Generalization bounds for downstream tasks:**
> >
> > For fine-tuning experiments, we consider two binary classification tasks on the QQP and Cola datasets from GLUE [2]. We contrast the bounds that we obtain for fine-tuning a pretrained GPT-2 small model [3], to training from scratch the same model on the two datasets. For both the pretrained and randomly initialized GPT-2 large models, we fine-tune using SubLoRA with rank = 8 and intrinsic_dim=30000 and perform fine-tuning. We run the fine-tuning for 5 epochs with a learning rate of 2e-5. We obtain the following non-vacuous classification accuracy bounds for both QQP and Cola:
> >
> > | Dataset | Error Bound for pretrained LLM (%) | Error Bound for trained from scratch LLM (%)   | Random Guess (%)  |
> > | ---       | ---          | ---        |  ---         |
> > | QQP     |    **35.27**      |     71.72      |     50       |
> > | Cola     |     **38.89**    |    53.42    |    50    |
> >
> > As we can see, using pretrained LLMs leads to tighter, non-vacuous bounds in contrast with training from scratch. Our bounds thereby provide a theoretical certification on the value of pretraining LLMs.
> > _______________
> > References:
> > [1]  Lotfi, S., Finzi, M., Kapoor, S., Potapczynski, A., Goldblum, M. and Wilson, A.G., 2022. PAC-bayes compression bounds so tight that they can explain generalization. Advances in Neural Information Processing Systems, 35, pp.31459-31473.
> >
> > [2] https://huggingface.co/datasets/glue
> >
> > [3] https://huggingface.co/gpt2

---

### Meta-Review · Area_Chair_GxHv · 2023-12-08

**Metareview:**

The paper aims to provide the generalization bounds for auto-regressive language models. The idea is interesting and the problem studied in this paper is highly important in the community. However, the reviewers think that this paper significantly combines existing techniques (e.g., SubLoRA), the novelty of this paper is very limited. and Reviewers also find that Theorem 1 can not be applied without independence assumption on X. These major issues can not be fixed in a shot time.

**Justification For Why Not Higher Score:**

N/A

**Justification For Why Not Lower Score:**

N/A

---

### Decision · Program_Chairs · 2024-01-16

Reject